# Fabrication and Performance of UV Photodetector of ZnO Nanorods Decorated with Al Nanoparticles

**DOI:** 10.3390/nano12213768

**Published:** 2022-10-26

**Authors:** Shiguang Shang, Yunpeng Dong, Wenqian Zhang, Wei Ren

**Affiliations:** School of Electronic Engineering, Xi’an University of Posts and Telecommunications, Xi’an 710121, China

**Keywords:** UV photodetector, ZnO nanorods, aluminum nanoparticle, hydrothermal method

## Abstract

In this work, localized surface plasmon resonance (LSPR) mediated by aluminum nanoparticles (Al NPs) was investigated to enhance the ultraviolet (UV) response of the zinc oxide nanorods (ZnO NRs) grown by the hydrothermal method. The ZnO NRs were characterized by scanning electron microscope, energy dispersive spectroscopy, X-ray diffractometer, Raman spectrometer, ultraviolet-visible spectrophotometer and fluorescence spectrometer. The results show that the morphology and crystalline structure of the ZnO NRs could not be changed before and after decoration with Al NPs, but the absorption rates in the UV range and the photoluminescence (PL) properties were improved. The photo-to-dark current ratio of ZnO NRs with Al NPs was about 447 for 325 nm UV light (5 mW/cm^2^) at 3.0 V bias, with the sensitivity increasing from 9.5 to 47.8, and the responsivity increasing from 53 to 267 mA/W.

## 1. Introduction

UV detection technology is a new technology used for both military and civilian requirements. It has a wide range of applications in space communications, flame detection, ozone monitoring, missile warning, etc. [1,2,3]. Many semiconductor materials, including GaN, TiO_2_, diamond, SiC and ZnO, have been developed and used in UV photodetectors [4,5,6]. Among them, ZnO semiconductor materials have attracted much attention due to their wide band gap, excellent chemical and thermal stability and specific electrical and optical properties with a large excitation binding energy (60 meV) [7]. In particular, one-dimensional NRs are getting increased attention due to their large surface-to-volume ratio, which can significantly reduce the reflection of UV light and enhance photon absorption [8,9,10], and for their good electron transportation capability by providing a direct conduction path for electron transport, reducing the number of grain boundaries greatly [11]. Thus, ZnO NRs are a significant material for high-sensitivity and fast-responsive photodetectors, resulting in high photoconductivity gain [12]. However, the ZnO NRs synthesized by the hydrothermal method contain a large number of surface defects and volume defects caused by unreacted organic residues and other chemical defects [13]. When ZnO NRs act as an electron transport layer in photoelectric devices, these defects can easily lead to the serious recombination of electrons and holes, increasing surface contact resistance and inducing low device performance [14]. To overcome the rapid recombination of electrons and holes, it was reported that surface decoration with metal NPs such as Ag, Au and Pd is an effective way to reduce the surface defects of ZnO and improve the device performance in the fields of optoelectronics and sensing [15,16,17]. In recent years, LSPR provided a novel idea for improving the performance of photoelectric detectors [17,18,19]; thus, it is of interest to contribute to enhance photodetection performance via the interaction between light and electron plasma waves at the metal particle surface. Moreover, the LSPR wavelength of metallic Al is similar to the intrinsic UV band edge emission of ZnO [16]. Compared to precious metal elements, the Al element is a good candidate for plasmonic material due to its abundance on earth and low costs. Recently, electron beam evaporation was employed to fabricate different material nanoparticles [20,21,22], which possesses advantages of compactness, uniformity and simple working conditions. Thus, a metal–semiconductor–metal (MSM) structured photodetectors of ZnO NRs decorated with Al NPs were fabricated by the hydrothermal method and electron beam evaporation, and the effect of Al NPs on the photoresponse performance of the ZnO NRs photodetectors was investigated.

## 2. Experimental Procedures

ZnO seed layer was deposited on quartz glass by RF magnetron sputtering using a ZnO target (99.999%). During the magnetron sputtering process, the working pressure was 1 Pa and the sputtering power was 120 W for 10 min. Then, the deposited ZnO seed layer was annealed in a furnace at 450 °C for 2 h. The aqueous chemical solution containing equal molar (0.05 molar) zinc acetate dehydrate (Zn(CH_3_COO)_2_) and hexamethylenetetramine (C_6_H_12_N_4_) was dissolved in 50 mL deionization water and stirred at room temperature for 30 min to ensure the complete mixing of the raw materials. The hydrothermal ZnO nanorod growth was carried out in a high-pressure reaction kettle by immersing the ZnO seed layer/quartz sample in the solutions and heating at 90 °C for 4 h. Particularly, the ZnO seed layer was faced downwards and leaned at about 45° against the reaction kettle in order to achieve a minimum temperature gradient growth effect. Al NPs (Al powder, purity 99.99%) were deposited on the surface of the ZnO NRs at 6 × 10^−4^ Pa by electron beam evaporation. The electron gun current was 280~320 mA and the deposition rate of Al NPs was 2 Å/s. The total thickness of the deposited Al NPs was 30 nm. To fabricate the MSM structure of conductive UV photodetectors, interdigited Ti film electrodes with a thickness of about 100 nm were deposited on ZnO NRs/quartz samples by DC magnetron sputtering.

The surface morphology of the ZnO NRs was characterized by scanning electron microscopy (SEM, JSM-6700F, Japan Electronics Co., Ltd., Tokyo, Japan) and the element composition of the ZnO NRs decorated with Al NPs was characterized by energy dispersive spectrometry (EDS, S-4800, Hitachi, Japan). The crystal structures of the ZnO NRs were determined by X-ray diffraction (XRD, 7000SAS, Shimazu, Japan) and Raman spectrometer (Raman, FI532W, Beijing Zhuoli Hanguang Instrument Co., Ltd., Beijing, China). The optical absorbance spectra of the ZnO NRs were measured by a UV-visible spectrophotometer (UV-2300Ⅱ, Shanghai Tianmei Scientific Instrument Co., Ltd., Shanghai, China). The PL spectra of the samples were measured by a fluorescence spectrometer (FluoroMax-4, HORIBA Scientific, Irvine, CA, USA). The current-voltage (*I*−*V*) characteristics and time response curves of the samples were measured using a multi-channel electrochemical workstation (CHI660E, Shanghai Chenhua Instrument Co., Ltd., Shanghai, China).

## 3. Results and Discussion

Figure 1 shows the SEM images of ZnO NRs decorated with and without Al NPs, and the EDS of ZnO NRs decorated with Al NPs. As shown in Figure 1a and the inset of Figure 1a, the ZnO NRs are a hexagonal columnar with good discrete structure, and the average diameter of the ZnO NRs is about 120 nm and their average length is about 2.5 μm. It can be seen from Figure 1b and the inset in Figure 1b that the ZnO NRs were decorated with Al NPs and Al NPs are attached to the top and side surface of the ZnO NRs. The element compositions Zn, O and Al of ZnO NRs decorated with Al NPs are shown in Figure 1c. The atomic percentages of Zn and O are 49.35 and 43.87%, respectively, and the atomic percentage of Al is only 6.78%. No other element peak is detected by EDS.

Figure 2a shows the XRD pattern of ZnO NRs decorated with and without Al NPs. It can be seen that at the diffraction angles 34.4°, 46.2° and 62.8°, the characteristic peaks of ZnO crystal planes (002), (102) and (103) appear, respectively, which is consistent with the standard spectrum (PDF#79-0206), indicating that the ZnO NR has a hexagonal wurtzite structure. The diffraction peak intensity of crystal plane (002) is strong and sharp, indicating that ZnO NRs possess high crystallinity along the c-axis orientation. No peaks attributable to the Al element were observed in the patterns of ZnO NRs decorated with Al NPs, which may be due to the low content of Al NPs.

Figure 2b shows the Raman spectra of ZnO NRs decorated with and without Al NPs. Raman scattering is an effective method to study the crystallization, structure and defects of nanomaterials. It can be seen from Figure 2b that the ZnO NRs decorated with and without Al NPs have a 2E_2_ (LO) mode scattering peak at the wavenumber 226 cm^−1^, which is related to the movement of zinc sublattice [23]. The scattering peak of the E_2_ (H) mode appears at the wavenumber 438 cm^−^^1^, indicating that the ZnO NRs decorated by Al NPs have high crystallinity [24] and confirming that the Al NPs do not affect the hexagonal wurtzite structure of the ZnO NRs. A weak broad peak belonging to the A_1_ (LO) mode appears at the wavenumber 574 cm^−^^1^, which is related to defects such as oxygen vacancies and zinc gaps in ZnO NRs [25]. 

Figure 3a shows the absorbance curve of ZnO NRs decorated with and without Al NPs. It can be seen that the ZnO NRs have strong absorption characteristics in the UV region within the 200−360 nm wavelength range, which is consistent with the inherent band gap width of ZnO semiconductors of 3.37 eV. The inset in Figure 3a is the Tauc plot obtained according to the absorbance curve of ZnO NRs, which means that the band gap of ZnO NRs decorated with Al NPs increases from 3.17 eV to 3.22 eV. Compared to pure ZnO NRs, the absorbance ratio of the ZnO NRs decorated with Al NPs increases from an average value of 99.64% to 99.93% in the wavelength range of 200~420 nm, indicating that the Al NPs can enhance the absorption efficiency of ZnO NRs in the UV range.

Figure 3b shows the PL spectra of ZnO NRs. It can be found that the ZnO NRs decorated with and without Al NPs have a strong PL violet emission peak near 384 nm, which is mainly associated with the recombination luminescence of free excitons, and the blue emission peak at 449 nm is caused by the zinc vacancy defects [26,27]. The blue-green emission peaks at 467 nm, 480 nm and 491 nm are related to oxygen vacancy, oxygen antisite defects, and so on [28,29]. Compared to pure ZnO NRs, the emission peak at 384 nm of ZnO NRs decorated with Al NPs blue-shifts to 378 nm, and the UV emission intensity increases more than two times, which can be attributed to the direct resonance coupling between the ZnO NRs excitons and the localized surface plasmons originated from the Al NPs. 

Figure 4a shows the *I*−*V* characteristics of the photodetector ZnO NRs decorated with and without Al NPs under dark and 325 nm light irradiation at a power density of 5 mW/cm^2^. It is found that the dark current of the UV photodetectors of ZnO NRs decorated with and without Al NPs are of the same order of magnitude, while the UV photodetector of ZnO NRs decorated with Al NPs performs better under 325 nm UV light irradiation, and the UV photodetector of ZnO NRs decorated with Al NPs has a higher photo-to-dark current ratio of about 447 at 3.0 V bias. Figure 4b shows the time-dependent photoresponse of ZnO NRs decorated with and without of Al NPs at zero bias under 325 nm light irradiation. It is found that the ZnO NRs show a stable and repeatable photocurrent response at zero bias. According to the definition [30], the sensitivity of the UV photodetector of ZnO NRs decorated with Al NPs is about 47.8, and that of pure ZnO NRs is about 9.5. Additionally, the calculated responsivities are 267 mA/W and 53 mA/W, respectively. 

Compared to the pure ZnO NRs, the photo-to-dark current ratio and the sensitivity and response speed of those decorated with Al NPs are improved. The main reason is attributed to LSPR producing Al NPs under UV light illumination. The electrons excited by LSPR have higher energy and will pass through the interface between Al NPs and Zn NRs and enter the internal ZnO NRs to conduct electricity [31,32]. The density of the conductive electrons and the Fermi level in the ZnO NRs also increase, which is confirmed by the phenomenon of the blue shift of the near-band edge emission peak in the PL spectra [18]. Additionally, due to the work function difference between ZnO NRs and Al NPs, a depletion region which gives a built-in electric field is developed at the interface under thermal equilibrium conditions. Under UV illumination, the photogenerated electron–hole pairs near the depletion region are separated by the built-in electric field and drift in opposite directions, resulting in the generation of photocurrent.

Figure 4c,d shows the enlarged rising and decaying edges of the transient photoresponse of the ZnO NRs decorated with Al NPs. The response time of 0.2 s and recovery time of 0.4 s of ZnO NRs decorated with Al NPs are obtained according to the definition of about 90% response time and 90% recovery time [3]. Using the same method, the response and recovery times of pure ZnO NRs are 0.4 s and 3.1 s, respectively. The results show that the ZnO NRs decorated with and without Al NPs all exhibit a fast response, while the former have a faster recovery time.

Figure 4e shows the photocurrents of the ZnO NRs decorated with Al NPs under different UV radiant intensities. By decreasing the radiant intensity, the carrier density in the ZnO NRs decreases accordingly, indicating that the photodetector of ZnO NRs decorated with Al NPs will have a lower photocurrent under weaker UV irradiation. The results suggest that the photocurrent of ZnO NRs decorated with Al NPs is a power function of UV light intensity.

## 4. Conclusions

In this work, ZnO NRs were fabricated by the hydrothermal method and were then deposited with Al NPs by electron beam evaporation. Al NPs can effectively enhance the absorbance of ZnO NRs under UV light irradiation and improve the separation of photo-induced electron–hole pairs. Compared to pure ZnO NRs, the UV photodetector made of ZnO NRs decorated with Al NPs exhibited a better performance with a photo-dark current ratio of about 447, a sensitivity of about 47.8, a responsivity of 267 mA/W and a response/recover time of less than 0.4 s. The results indicate that ZnO NRs decorated with Al NPs are promising materials for the future generation of UV photodetectors.

## Figures and Tables

**Figure 1 nanomaterials-12-03768-f001:**
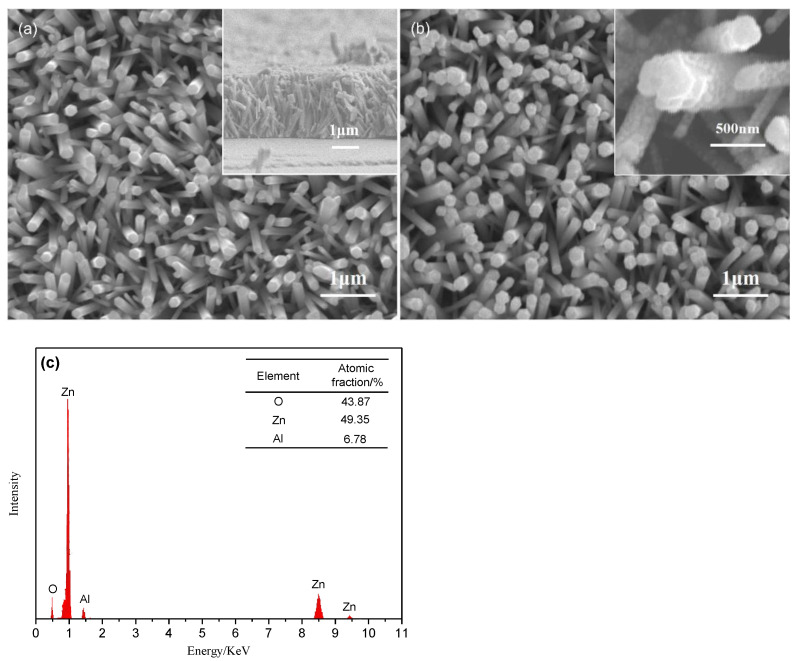
FE-SEM images of ZnO NRs decorated without (**a**) and with (**b**) Al NPs, and EDS of ZnO NRs decorated with Al NPs (**c**).

**Figure 2 nanomaterials-12-03768-f002:**
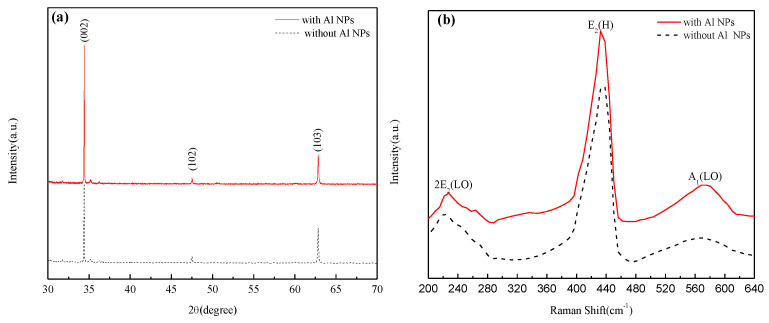
XRD spectra (**a**) and Raman spectra (**b**) of ZnO NRs decorated with and without Al NPs.

**Figure 3 nanomaterials-12-03768-f003:**
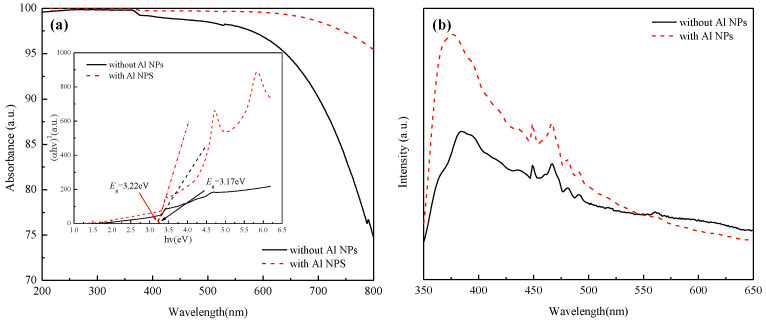
Absorbance spectra (**a**) and PL spectra (**b**) of ZnO NRs decorated with and without Al NPs.

**Figure 4 nanomaterials-12-03768-f004:**
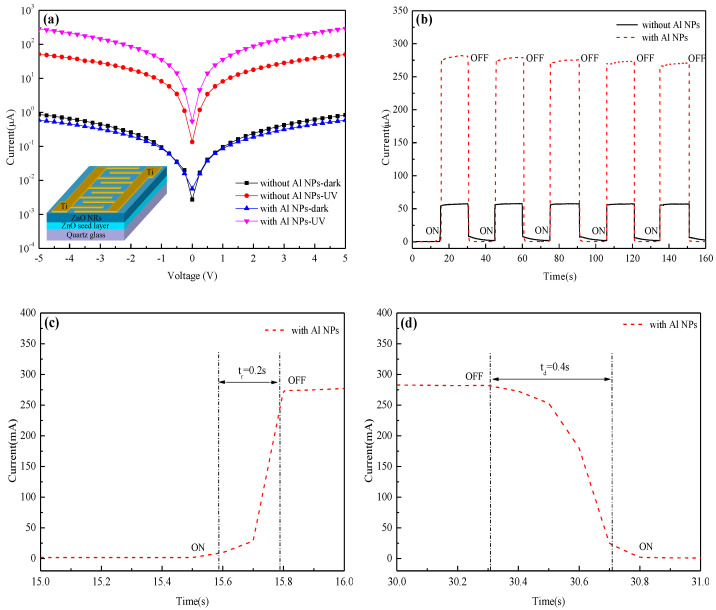
*I*−*V* characteristics of the photodetector of ZnO NRs (**a**), time-dependent photoresponse of ZnO NRs (**b**), enlarged rising and (**c**) decaying edges (**d**) of the transient photoresponse of ZnO NRs measured by pulse UV irradiation, and *I*−*V* characteristics of ZnO NRs decorated with Al NPs under different UV radiant intensities (**e**).

## Data Availability

The data in the article is valid.

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
