# Peer review of "Fabrication and Performance of UV Photodetector of ZnO Nanorods Decorated with Al Nanoparticles"

_nanomaterials, 2022, doi:10.3390/nano12213768_

Round 1

Author Response

Dear reviewer,
Thank you very much for your positive comments and suggestions on our article! From your positive comments and suggestions, I can feel that you are an expert in this field, and your suggestions are deep and delicate, which will greatly improve us.

Please refer to the attachment for specific reply!

Thanks!
Best Regards!
Shiguang Shang
Email:[email protected]

Reviewer 2 Report

This paper by Shang et al. has considerably improved the ZnO nano road-based UV photodetector performance using Aluminum nanoparticles. The reviewer supports the publication, with following comments and suggestions.

·         The authors stated in the abstract about studying LSPR; however, there are no experimental measurements of LSPR in their work. They should provide an explanation and experimental measurements of it.

·         Authors claimed to obtain the Al nanoparticle by e-beam deposition; however, there is no experimental evidence of Al nanoparticles in the work. Also, authors should reference where people obtained the nanoparticle by e-beam deposition.

·         In Fig 3a ), providing an absorption spectrum would be better than the transmission and obtaining the actual bandgap from it.

·         In the PL paragraph from line 125, they should explain more clearly what they mean by gap and whether they are referring to defects or other gaps. And why is the main emission peak relatively broader, unlike narrow (or sharper) in literature?

·         According to the claim of the UV photodetector, the Authors should provide the figure of merit parameters such as responsivity, detectivity, and photoresponse speed (Surfaces and Interfaces 23 (2021) 100934).

·         ZnO is a crucial material for the development of energy-efficient UV-based optoelectronics. Authors should highlight this significant merit to provide impactful information to readers. Suggested references are Adv. Electron.Mater.2019, 5, 1900348; Nano Energy 91 (2022) 10667; Nano Energy 90 (2021) 106496; Cell Reports Physical Science 2 (2021) 100591; Solar Energy Materials and Solar Cells 194 (2019) 148–158; Nanoscale Adv., 2019,1, 2059-2085; Nanomaterials 2020, 10, 395;

Author Response

(The authors gave the same response as above.)

Round 2

Reviewer 1 Report

The revised manuscript of "Fabrication and Performance of UV Photodetector of ZnO Nanorods Decorated with Al Nanoparticles" shows a well-considered study and provides valuable information.

Reviewer 2 Report

Authors have improved the revised version and reviewer support the publication of the work